# Single crystal of a one-dimensional metallo-covalent organic framework

Hai-Sen Xu [1,7], Yi Luo [2,3,7], Xing Li [1,7], Pei Zhen See[1], Zhongxin Chen[1], Tianqiong Ma[2], Lin Liang[4], Kai Leng [1], Ibrahim Abdelwahab [1], Lin Wang [1], Runlai Li[1], Xiangyan Shi[5], Yi Zhou[6], Xiu Fang Lu[1], Xiaoxu Zhao[1], Cuibo Liu[1], Junliang Sun [2✉] & Kian Ping Loh [1✉]

Although polymers have been studied for well over a century, there are few examples of covalently linked polymer crystals synthesised directly from solution. One-dimensional (1D) covalent polymers that are packed into a framework structure can be viewed as a 1D covalent organic framework (COF), but making a single crystal of this has been elusive. Herein, by combining labile metal coordination and dynamic covalent chemistry, we discover a strategy to synthesise single-crystal metallo-COFs under solvothermal conditions. The single-crystal structure is rigorously solved using single-crystal electron diffraction technique. The non-centrosymmetric metallo-COF allows second harmonic generation. Due to the presence of syntactic pendant amine groups along the polymer chains, the metallopolymer crystal can be further cross-linked into a crystalline woven network.

[1] Department of Chemistry, National University of Singapore, 3 Science Drive 3, Singapore 117543, Singapore. [2] College of Chemistry and Molecular Engineering, Beijing National Laboratory for Molecular Sciences, Peking University, 100871 Beijing, China. [3] Department of Materials and Environmental Chemistry Stockholm University, SE-10691 Stockholm, Sweden. [4] State Key Laboratory of Applied Organic Chemistry, College of Chemistry and Chemical Engineering, Lanzhou University, 730000 Lanzhou, Gansu, China. [5] School of Physical and Mathematical Sciences, Nanyang Technological University, 21 Nanyang Link, Singapore 637371, Singapore. [6] School of Physical Science and Technology, Shanghai Tech University, 201210 Shanghai, China. [7] These authors contributed equally: Hai-Sen Xu, Yi Luo, Xing Li. ✉email: junliang.sun@pku.edu.cn; chmlohkp@nus.edu.sg

Framework solids have been one of the hottest materials over the past 30 years because they represent mankind's attempt to control chemical bonding in space versus random polymerisation[1,2]. Depending on the nature of the strongest bond (coordination bond, covalent bond, hydrogen bond, etc.) used in constructing the solids, framework materials are categorised into metal-organic frameworks (MOFs)[3–5], covalent organic frameworks (COFs)[6–10], and hydrogen-bonded organic frameworks (HOFs)[11,12]. The synthesis of framework materials needs to be carried out under conditions where the bond formation is highly reversible to facilitate the self-correction process necessary for crystal growth. In this regard, COFs are among the most difficult to crystallise, owing to the lesser reversibility of their covalent linkages compared to coordination bonds and hydrogen bonds in MOFs and HOFs. The ease of encoding functionalities in COFs and their structural robustness render them potentially useful in wide-ranging applications[13–19]. However, an in-depth understanding of the structure-property correlation in COFs is lacking, owing to the fact that most synthesised COFs are polycrystalline, which hampers structural determination. Recently, the addition of crystal seeds or modulators have been used to grow single-crystalline two- or three-dimensional (2D or 3D) COFs[20,21]. Nevertheless, only a few examples of 3D COFs have their crystal structure rigorously solved[21–23]. It is generally recognised that single-crystalline frameworks of lower dimensionality are more difficult to grow compared to higher dimensionality ones[24].

Although the vast choices of COF building units and covalent linkages give rise to diverse structural motifs in the synthesised product, the produced 2D and 3D COFs tend to crystallise in high symmetric space groups due to the high symmetries of the building blocks and low freedom of intermolecular packing[8,9]. As such, conventionally synthesised COFs are mostly centrosymmetric and do not exhibit second harmonic generation or ferroelectricity[25]. In this regard, one-dimensional (1D) COFs, which possess a high degree of freedom in molecular packing, can be a candidate to construct non-centrosymmetric crystals. Conceptually, the framework structure in 1D COF is constructed from 1D-confined covalent linkages, and non-covalent interactions (such as π-π interactions, hydrogen bonding, etc.) in the other two dimensions help to pack the 1D chains. However, the extremely high anisotropy and the entropy-driven random packing of organic chains impose a huge challenge on the synthesis of 1D COFs. Using its covalent analogue, linear polymers, as examples: since the invention of the first synthetic polymer (Bakelite) in 1907, nearly all the polymers synthesised are amorphous or poorly crystalline. To obtain a well-defined 1D COF, two basic aspects have to be considered: (i) How to get a framework, not just a densely packed organic polymer; (ii) how to control the periodic packing of organic chains to get a single-crystalline structure. Although the topochemical polymerisation approach has been used to produce single-crystalline polymers from pre-packed molecular crystals, the approach is rather limited in scope[26–28]. It is highly desirable to search for a direct crystallisation method from solution, which has been so far elusive.

Herein, we demonstrate a strategy to synthesise single-crystalline 1D metallo-COFs by combining metal-ligand coordination[29] and dynamic covalent chemistry (DCC)[30] (Fig. 1). A one-pot reaction combining self-assembly[31,32] and imine condensation is conducted under solvothermal conditions. Along with the poly-condensation reaction, the constituents are consumed and regenerated from the reactant "pool"[33]. An advantage of this method is that building blocks can evolve into multi-intermediates through reversible reactions or interactions without the need to synthesise each building block individually, which is distinct from the conventional methods of making crystalline organic networks using predetermined building blocks[34,35].

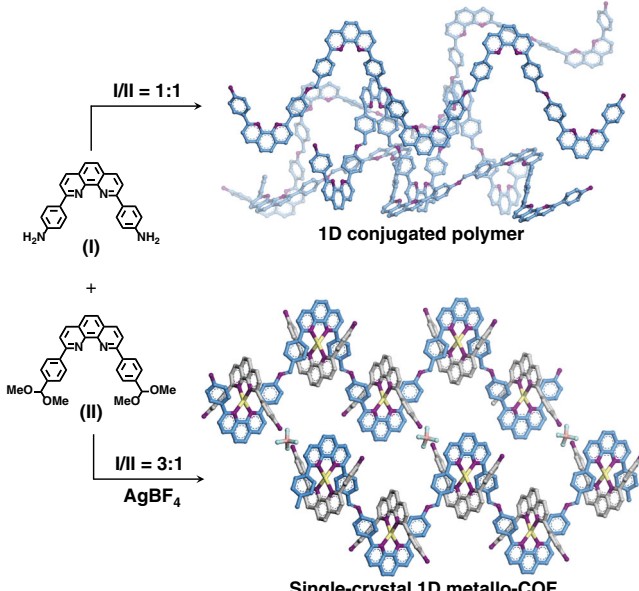

**Fig. 1 Strategy of synthesising single-crystalline 1D metallo-COF.** Colour scheme: C on organic chains, blue; C on pendants, grey; N, purple; Ag, yellow; B, pink; F, green. Hydrogen atoms are omitted for clarity.

## Results

**Synthesis of mCOF-Ag**. We first attempt to construct 1D conjugated polymer solely based on DCC without a metal template. Due to the absence of secondary interactions, the 1D polymer chains pack randomly to afford a poorly crystalline polymer (Supplementary Fig. 1). We then introduce AgBF$_4$[36] to initiate ligand exchange and provide an additional reversible process for the poly-condensation, which should be beneficial for the crystal nucleation process. When one of the building units, 4,4′-(1,10-phenanthroline-2,9-diyl)dianiline (**I**), is in excessive amount, it can be anchored onto the backbone chains via Ag$^I$ coordination in a syntactic fashion, thus providing periodic spacers to induce π–π stacking and hydrogen bonding between the 1D chains (Fig. 1). Using this approach, we synthesise a single-crystalline 1D metallo-COF (mCOF-Ag) with micrometre-sized particles, allowing the rigorous characterisation of the crystal structure via single-crystal electron diffraction (SCED)[37].

In line with the strategy, a one-pot reaction by combining self-assembly and imine condensation was conducted under solvothermal conditions. In 2,9-bis(4-(dimethoxymethyl)phenyl)-1,10-phenanthroline (**II**), an acetal group[38] is preferred for its good solubility which is beneficial for the self-assembly process; besides, the relatively low reactivity of acetal with amine groups can reduce the crystal nucleation rate to enable good crystals. Once AgBF$_4$ was added into the suspension of **I** and **II**, an immediate colour change was observed indicating the occurrence of self-assembly via coordination. The amount of AgBF$_4$ added has a strong influence on crystal qualities. According to the strategy, the stoichiometry of **I**, **II**, and AgBF$_4$ is 3:1:2. However, adopting this ratio produces a large number of Ag particles on the material. The presence of Ag particles is attributed to the reversible nature of imine bond; in situ formed aldehyde groups can reduce Ag ions into elemental Ag, which then aggregated into particles. When the ratio of AgBF$_4$ is reduced to 1, no Ag particles were detected under the same reaction conditions. The use of excess **I** and **II** can protect Ag ions from being reduced. After carefully screening the reaction conditions, mCOF-Ag was obtained in a mixture of 1-butanol, 1,2-dichlorobenzene, and 6 M aqueous acetic acid (1/9/1, v/v/v) at

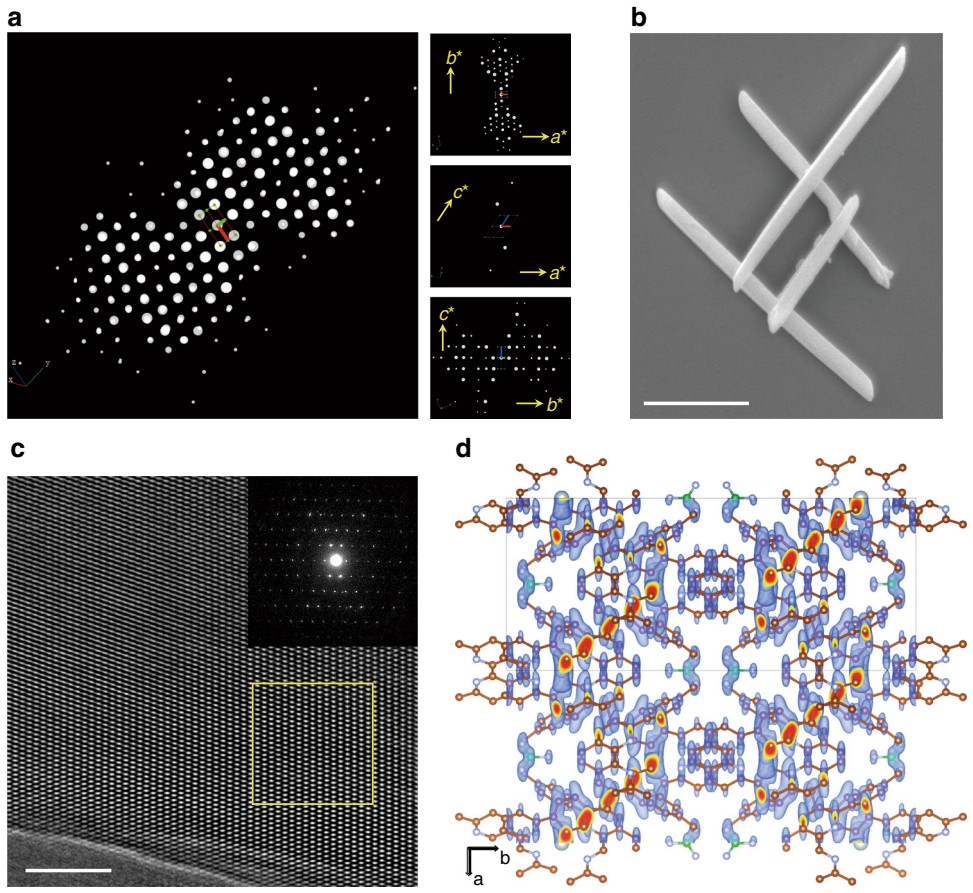

**Fig. 2 Structure characterisation and analysis of mCOF-Ag. a** 3D reciprocal lattice of mCOF-Ag reconstructed from the SCED data (left) and 2D slices cut from the reconstructed reciprocal lattice (right). **b** SEM image of mCOF-Ag with uniform rod-like morphology. Scale bar: 1 μm. **c** HRTEM image of mCOF-Ag and SAED pattern (inset) confirmed the single-crystal nature of the material. Scale bar: 20 nm. **d** Observed electron density map of the initial structure model determined from SCED data along the c-axis.

120 °C for three days with a feeding ratio of **I**, **II**, and $AgBF_4$ equals to 3:1:1. After copiously washing with dimethyl sulfoxide (DMSO) and tetrahydrofuran (THF), and drying at 120 °C for 8 h, mCOF-Ag was obtained as a light yellow solid (74% yield, based on $AgBF_4$).

**Characterisation.** mCOF-Ag exhibits sharp powder X-ray diffraction (PXRD) peaks, indicating its high crystallinity (Supplementary Fig. 2). The chemical bonding in mCOF-Ag was assessed by Fourier-transform infra-red spectroscopy (FT-IR) (Supplementary Figs. 3 and 4) and[13] C cross-polarisation magic-angle spinning (CP/MAS) NMR (Supplementary Fig. 5). Scanning electron microscopy (SEM) image reveals that mCOF-Ag exhibits a uniform rod-like morphology with the crystal size >2 μm (Fig. 2b). Ordered lattice fringes in the high-resolution transmission electron microscopy (HRTEM) image (Fig. 2c) and selected-area electron diffraction (SAED) pattern (Fig. 2c, inset) confirm its single-crystal nature.

The single-crystal structure of mCOF-Ag is determined directly from the SCED data[39,40] (Fig. 2a). From the observed electron density maps (Fig. 2d and Supplementary Fig. 8), all positions of the non-hydrogen atoms (C, N, and Ag) on the framework structure and the locations of the guests ($BF_4^-$ anions) are identified. The SCED data was collected on a typical rod-like crystal, with a resolution of ~0.95 Å, which reflects the single-crystal nature of mCOF-Ag. A C-centered monoclinic cell with $a = 15.66$ Å, $b = 31.00$ Å, $c = 10.87$ Å, and $\beta = 123.31°$ is identified by indexing the SCED data using the programme XDS

(Supplementary Table 1). As indicated by the reflection conditions (hkl: $h + k$; 0kl: $k = 2n$; hk0: $h + k$; h00: $h = 2n$; and 0k0: $k = 2n$), the possible space groups are deduced to be C2/c (No.15), Cc (No. 9), C2/m (No. 12), Cm (No. 8), or C2 (No. 5). Based on this SCED data, the structure model of mCOF-Ag was solved directly in the space group C2/c using the programme SHELXT (dual-space method). In the determined structure model, it is found that the C and N atoms on the connection bonds (–C = N–) are symmetry-related. In order to make them symmetry independent, the symmetry of the structure model was reduced into its subgroup C2. The determined structure model of mCOF-Ag was further confirmed and refined against its synchrotron powder X-ray diffraction (SPXD) data (Supplementary Fig. 11). Using the SVD-index method implemented in the software Topas, the SPXD data of mCOF-Ag is indexed by a C-centered monoclinic cell of $a = 15.83$ Å, $b = 29.97$ Å, $c = 10.69$ Å, and $\beta = 123.96°$, which is consistent with the unit cell obtained from the SCED data. The final agreement residuals for the Rietveld refinement are $R_I = 0.075$, $R_{wp} = 0.108$, with $R_{exp} = 0.049$.

Further analysis of the crystal structure reveals more insights about the packing of the 1D chains. The adjacent phenanthroline chains are arranged in a parallel fashion to generate a 2D corrugated layer (Supplementary Fig. 10), and the 2D layers are further stabilised by the interactions of amines with $BF_4^-$ anions (Supplementary Fig. 9), as well as the π–π stacking of interlayer phenanthroline rings (Supplementary Fig. 10), to form a permanent structure (Fig. 3a). The Ag and pendant groups **I**

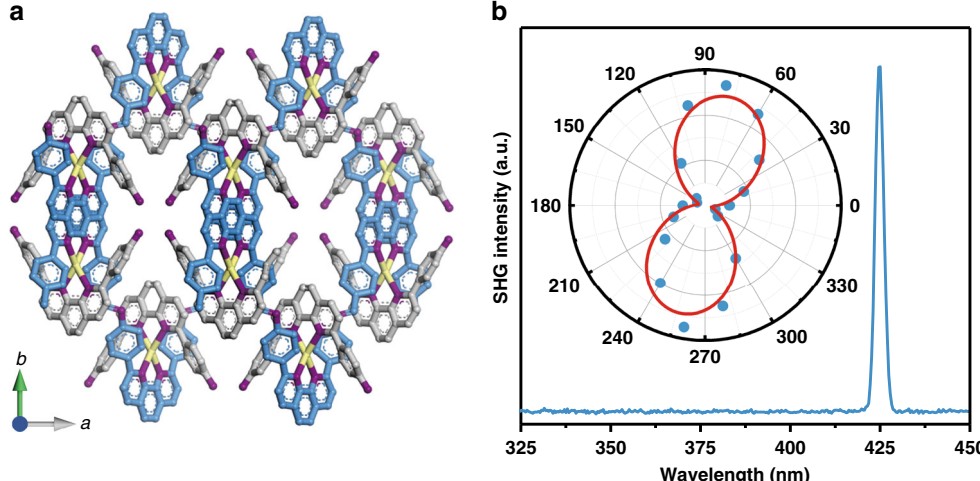

**Fig. 3 Single-crystal structure and nonlinear optical measurements of mCOF-Ag. a** Crystal view of mCOF-Ag along the c-axis. Colour scheme: C on organic chains, blue; C on pendants, grey; N, purple; Ag, yellow. Hydrogen atoms and $BF_4^-$ anions are omitted for clarity. **b** The SHG spectrum of an isolated crystal of mCOF-Ag. The excitation wavelength is 850 nm, the peak at 425 nm is the SHG signal. Inset: polarisation dependent SHG response. Red curve is the fitting result of experimental data (blue dots). The units are in degree.

are arranged in a syntactic version, favouring ordered packing of the polymer chains. mCOF-Ag belongs to the C2 space group, which is non-centrosymmetric. This is validated by second harmonic generation (SHG)[41] at 425 nm when the crystal is excited by a fundamental laser wavelength of 850 nm (Fig. 3b and Supplementary Fig. 16). Polarisation-dependent SHG was recorded on an isolated crystal. Using polarised laser excitation, a SHG map of the COF-Ag crystal could be obtained (Fig. 3b, inset). The strongest SHG response was observed under parallel polarised excitation, which means that the maximum SHG response originates from the longitudinal direction of the mCOF-Ag crystal.

**Crystalline state polymerisation.** Due to the packing of zigzag chains, the orientation and rotation of amine groups in building block **I** are spatially well defined and the two diagonally positioned –NH$_2$ groups are separated by ~3.9 Å (Fig. 4a). mCOF-Ag is an ideal scaffold for crystalline-state polymerisation because the adjacent amine groups can be linked by bi-functional compounds. Glyoxal, a molecule with two aldehyde groups, was chosen for its suitable molecular size and good reactivity with amine groups. mCOF-Ag was reacted with glyoxal solution (40 wt.% in H$_2$O) at 70 °C in 1,4-dioxane. After incubation for 5 days and washing, the product was collected and dried to yield a cross-linked woven network, termed as wCOF-Ag (Fig. 4a). The crystalline-state polymerisation, which occurs via "stitching" of the inter-chain amine groups by the aldehyde molecules, was verified using a range of techniques. In the FT-IR spectra, the characteristic bands at 3361 cm$^{-1}$, 3301 cm$^{-1}$, and 3197 cm$^{-1}$ for N–H stretching, and at 1618 cm$^{-1}$ for N–H bending, have almost vanished after cross-linking, indicating the near-complete consumption of amine groups in mCOF-Ag (Supplementary Fig. 17). The polymerisation process was further verified by the CP/MAS NMR spectrum of wCOF-Ag, where the peak corresponding to the aldehyde carbon in glyoxal at 190 ppm is absent (Supplementary Fig. 18), indicating the full connection of amine and aldehyde groups. The PXRD pattern appears very similar to mCOF-Ag except for slight differences at high angles (Fig. 4b). These results affirm that the cross-linking process did not affect the major part of the crystal structure. One evidence that the cross-linked solid is different from the pristine structure is provided by the photoluminescence (PL) spectroscopy, where the PL

of wCOF-Ag showed a dramatic sixfold enhancement in intensity compared to that of mCOF-Ag (Supplementary Fig. 23). We have also examined the mechanical properties of the crystals before and after cross-linking using nano-indentation. By indenting on an isolated micron-sized crystal of each of these two samples, Young's moduli of mCOF-Ag and wCOF-Ag were determined to be 9.0 (Supplementary Fig. 24) and 19.1 GPa (Supplementary Fig. 25), respectively (Fig. 4c). The value of wCOF-Ag is comparable with that of COF-505 (~12.5 GPa)[34]. The distinct increase in Young's modulus of wCOF-Ag is in line with its covalently connected structure in a 3D space, which imbues greater rigidity on the resulting material.

## Discussion

In this study, by combining metal-ligand coordination and dynamic imine bond formation, a single-crystalline material, mCOF-Ag, was successfully constructed via solvothermal conditions; its crystal structure was rigorously solved with SCED. Due to its non-centrosymmetric structure, mCOF-Ag shows an obvious SHG signal, demonstrating its potential as nonlinear optical materials. Moreover, due to the presence of interlaced pendant amine groups along the polymer chains, a high degree of control on the polymer backbone is obtained by crystalline-state polymerisation to form a woven network. From a synthetic perspective, the strategy we developed combines the powerful templating effect of metal ions and cooperative organisation of building blocks to maximise the number of secondary bonds in solution; thus it can be a promising method for the solution-phase synthesis of single-crystal covalent metallopolymers with unique topologies and functionalities.

## Methods

**Crystallisation of single-crystalline mCOF-Ag.** **I** (17.4 mg, 0.048 mmol) and **II** (7.7 mg, 0.016 mmol) were weighed into a 10 mL Schlenk tube. To the mixture, a solution of AgBF$_4$ (3.1 mg, 0.016 mmol) in 1-butanol (0.1 mL) and 1,2-dichlorobenzene (0.9 mL) was added. After sonicating at room temperature for 10 min, 0.1 mL 6 M aqueous acetic acid was added into the suspension. The Schlenk tube was frozen in a liquid nitrogen bath, evacuated to an internal pressure of 0 mbar and sealed. After warming to room temperature, the Schlenk tube was placed into an oven and heated at 120 °C for 3 days yielding a light yellow solid at the bottom of the tube. The Schlenk tube was opened when the mixture was still warm, and the solid was transferred into a vial, separated by centrifugation, washed copiously with DMSO and THF. After drying at room temperature, the resulting solid was dried under vacuum at 120 °C for 8 h to obtain mCOF-Ag as a light yellow solid

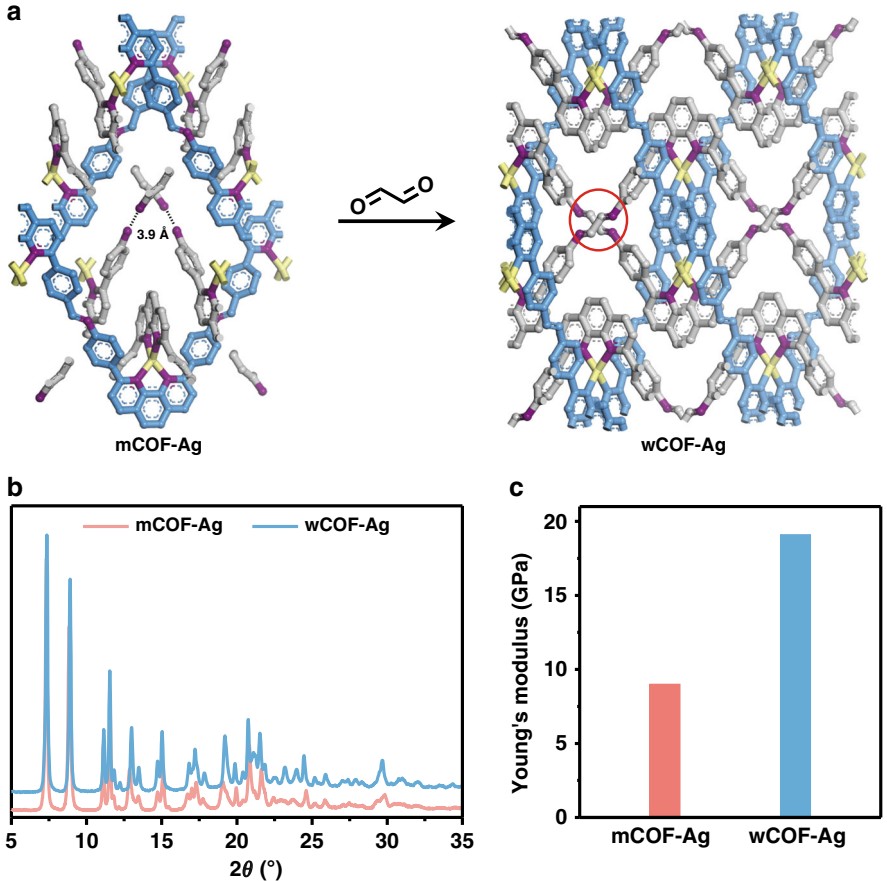

**Fig. 4 Crystalline-state polymerisation and characterisation. a** Confined environments of mCOF-Ag (left) and the ideal structure of the cross-linked framework, wCOF-Ag (right). One of the cross-linked parts is highlighted with a red circle. Colour scheme: C on organic chains, blue; C on pendants, grey; N, purple; Ag, yellow. Hydrogen atoms and $BF_4^-$ anions are omitted for clarity. **b** PXRD patterns of mCOF-Ag and wCOF-Ag. **c** The Young's moduli of mCOF-Ag and wCOF-Ag.

(10.8 mg, 74% yield, based on $AgBF_4$). Anal. Calcd for $(C_{49}H_{33}N_7AgBF_4)_n$: C 64.35; H 3.64; N 10.72. Found: C 64.29; H 3.61; N 10.67.

**Crystalline state polymerisation**. To a 10-mL pressure tube were added mCOF-Ag (5.0 mg), 1,4-dioxane (1.0 mL), and glyoxal solution (40 wt.% in $H_2O$) (5.0 μL). The tube was sealed and placed in an oven at 70 °C for 5 days. After cooling to room temperature, the solid was transferred into a vial and washed with THF, $H_2O$, EtOH, and $Et_2O$ in sequence. The resulting solid was dried at room temperature to afford the polymerised material, wCOF-Ag. Anal. Calcd for $(C_{51}H_{31}N_7AgBF_4)_n$: C 65.41; H 3.34; N 10.47. Found: C 65.92; H 3.26; N 10.35. Temperature is an important factor to obtain a high-quality material. Increasing the temperature from 70 °C to 80 °C will generate Ag particles on the surface of the crystal. The Ag particles are attributed to the labile nature of Ag ions, which can be reduced to elemental Ag by glyoxal at a higher temperature.

## Data availability

The single-crystal structure of mCOF-Ag is archived at the Cambridge Crystallographic Data Centre under the reference number CCDC-1956747. This data can be obtained free of charge from The Cambridge Crystallographic Data Centre via www.ccdc.cam.ac.uk/data_request/cif. All data are available from the authors upon reasonable request.

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

## Acknowledgements
K.P.L. acknowledges NRF-CRP grant "Two Dimensional Covalent Organic Framework: Synthesis and Applications". Grant number NRF-CRP16-2015-02, funded by National Research Foundation, Prime Minister's Office, Singapore. J.S. and Y.L. acknowledge National Natural Science Foundation of China (21527803 and 21871009) and Swedish Research Council. We thank Qi Zhang (National University of Singapore) for help with PL tests, Dr. Rodrigo Berte, Prof. Stefan A. Maier (LMU Munich), Ziyu Zhu, Xiao Wu, and Qiangbing Guo (National University of Singapore) for help with SHG measurements, and Lei Zhang, Pohua Chen (Peking University) for help with single-crystal structure analysis.

## Author contributions
K.P.L. supervised the project. J.S. supervised the crystal structures analysis. H.-S.X. designed and performed the experiments. Y.L. conducted SCED characterisation and Rietveld Refinement. X.L., Z.C., T.M., L.L., K.L., X.F.L., and C.L. discussed the synthesis and characterisation. I.A. conducted the SHG test. L.W. conducted the AFM test. R.L. helped to analyse the nano-indentation data. X.S. conducted CP/MAS NMR characterisation. Y.Z. conducted HRTEM characterisation. X.Z. helped to analyse the HRTEM data. K.P.L., H.-S.X., X.L., P.Z.S., and Y.L. wrote the manuscript.

## Competing interests
The authors declare no competing interests.
