## [Peer Review File · Nature Communications]

Reviewers' comments:

Reviewer #1 (Remarks to the Author):

In their excellent current report, Junliang Sun, and Kian Ping Loh et al., have proposed an idea to develop new Single Crystal of a One-dimensional Covalent Organic Framework. The study presents an interesting class of materials which are well characterized through many techniques to describe their structure and chemical environment. Hence, I recommend acceptance per minor revision. Below are a few points that should be considered prior to publication.

1. I wonder if there is much scope for bulk phase materials synthesis [J. Am. Chem. Soc., 2018, 141, 1807 and J. Am. Chem. Soc., 2017, 139, 1856-1862] of this 1-D COFs considering low yield, crystallinity and porosity. Would love to hear about authors thought on this aspect.
2. Maybe I have missed it, but I could not find a solid-state NMR and PXRD data and other electron diffraction data of wCOF-Ag. It should be provided.
3. The IR data of wCOF-Ag should be explained a bit more and so as the PXRD data.

Reviewer #2 (Remarks to the Author):

This work shows a novel and well-designed synthetic approach to produce single crystals of a 1D-metallo COF using a combination of two different dynamic bonds, coordination bonds and covalent bonds. The structure of the new COF has been successfully solved using single-crystal electron diffraction. Additionally, the initial m-COF-Ag network is further polymerized to form a novel woven 3D network. The results presented are very important suggesting a new way to obtain 1D COF single crystals in a controlled manner. The manuscript is well-written and deserves publication with very minor corrections.

Minor questions:

- 1) What happen when the metal center is removed from mCOF-Ag and wCOF-Ag?
- 2) In order to corroborate the coordination environment of the Ag(I) after the conversion from mCOF-Ag to wCOF-Ag, for instance XPS analysis would be interesting.
- 3) Have the authors analyze the adsorption properties of these 3 new materials?

Reviewer #3 (Remarks to the Author):

The manuscript describes the synthesis of a metallated 1-D covalent organic framework - a polymer with

a regular connectivity that packs in an ordered solid state structure - with a single crystalline morphology, and its subsequent cross-linking to form a woven network. The initial framework is structurally characterized by single crystal electron diffraction and its non-centrosymmetric structure is confirmed by the observation of non-linear optical properties. Nanoindentation measurements show the enhanced mechanical properties of the woven material. The preparation of single crystalline COFs is rare, and, as the authors printout, particularly so for low-dimensional materials.

As far as I am aware, this is the first report of a 1-D single crystalline COF, albeit a metallated one and that requires electron diffraction, rather than X-rays to fully characterize the structure. Despite those caveats, the elegance of the synthetic method using reversible chemistry and the metal template to form a highly ordered structure combined with the complexity of the structure make this an article that will be of broad interest to researcher working in the very active areas of framework solids and network materials more widely. The use of templates has obviously been used in diverse areas of chemistry, but given the current interest in preparing highly crystalline COFs, particularly as reports of COFs with varied functional properties are proliferating, methods to produce single or even highly ordered polycrystalline solids are essential. This work adds to the those possible tools and facilitating the use of crystallographic methods for COFs will facilitate reliably determination of the structures of COFs in order to better elucidate structure-function relationships in this class of materials. As an aside, I wonder if this material could have been determined directly from synchrotron XRPD based on an idealized structural model as a starting point.

The manuscript is well written; it is succinct and the text presents the results and supporting evidence very clearly. I would support the publication of this manuscript in Nature Communications.

There are a few minor revisions and comments that I would encourage the authors to address before publication.

- The main issue is presentational: The figures of the crystal structures are extremely difficult to understand, even viewing them alongside the descriptions in the text. For example, the pi-pi stacking and hydrogen bonding between chains are invoked in the text, but not shown in the figures; the packing diagrams (Fig. 1, 3a) do not show the crystallographic axes and in Fig. 2d, the electron density makes it hard to see the atoms, so overall, it is difficult to visualize the complete 3-D structure without opening the cif in a visualizer. The figures in the ESI do not help significantly. It is very difficult to see how the 'corrugated layers' in Fig. S8 are corrugated and to understand how the other representations of the structure relate to this. I would suggest having a single figure specifically for a simple representation of the structure, even as a schematic, to show some of the higher order features, such as the layers, the strand/chains that make up the structure. The structure itself is beautifully complex and it would be a shame if the authors cannot present it in a clear way.
- Do the authors have any idea why is the conjugated polymer crystalline? Is this a fully polymeric material or is there any previous literature that would lead them to expect it to be crystalline? A comment in the ESI would suffice.
- I would encourage the authors to provide a little more information on the treatment of the COF in

terms of drying/activation. The framework is washed with DMSO, THF, EtOH. Is it confirmed that no solvent is retained in the structure? The TGA shows significant mass loss before the apparent onset of decomposition at approx. 320 °C - what is being lost?

- I would also suggest a slightly fuller description of the polarized SHG measurement, with in terms of what the experiment actually is and what the results show. It is not a common technique in this area.
- The authors should present X-ray powder profile simulated from the single crystal cif vs. the experimental XRPD to see how different/similar the bulk microcrystalline powder is from the determined single crystal structure
- Have the authors tried Rietveld refinement for the woven COF. The apparent similarity of the two materials would suggest the mCOF-Ag would be a reasonable basis for a starting model. Even if Rietveld is not feasible at least Pawley/Le Bail refinement for wCOF-Ag should be included.

Point-by-point Response Letter

Ref.: “Single Crystal of a One-dimensional Covalent Organic Framework”, by Xu Hai-Sen *et al.* NCOMMS-19-37619-T.

[The original comments offered by the reviewers are quoted in *Italics* and our responses are described in blue.]

=====

Response to reviewer 1:

Reviewer #1 (Remarks to the Author):

In their excellent current report, Junliang Sun, and Kian Ping Loh et al., have proposed an idea to develop new Single Crystal of a One-dimensional Covalent Organic Framework. The study presents an interesting class of materials which are well characterized through many techniques to describe their structure and chemical environment. Hence, I recommend acceptance per minor revision. Below are a few points that should be considered prior to publication.

We thank the reviewer for his/her positive comments.

1. I wonder if there is much scope for bulk phase materials synthesis [J. Am. Chem. Soc., 2018, 141, 1807 and J. Am. Chem. Soc., 2017, 139, 1856-1862] of this 1-D COFs considering low yield, crystallinity and porosity. Would love to hear about authors thought on this aspect.

[Reply]: Thank you for your value suggestion. Bulk phase materials synthesis is one of the bottlenecks in the synthesis of COF. The co-agent assisted solid state strategy developed by Banerjee and co-workers is a very promising method indeed, but unfortunately, we didn't manage to get it to work in this case. The failure to synthesize mCOF-Ag by applying the solid state synthetic strategy is due to the following reasons: (1) the building block II is an acetal-protected compound, which has low reactivity with amine functionalized building block I; (2) in our system, a dynamic self-assembly occurs under the solvothermal conditions and the Ag phenanthroline complexes formed in situ are necessary for constructing mCOF-Ag. Dynamic self-assembly is difficult to occur in the solid state.

We have tried to follow the procedures in *J. Am. Chem. Soc.*, **139**, 1856-1862 (2017) to

synthesize mCOF-Ag in solid state. *p*-Toluenesulfonic acid monohydrate (1.0 mmol, 190.2 mg) was placed in a mortar followed by the addition of building block I (0.18 mmol, 65.2 mg). The reaction mixture was thoroughly ground with a pestle for 5 minutes. Then building block II (0.06 mmol, 28.8 mg) and AgBF₄ (0.06 mmol, 11.7 mg) were added to the mixture and mixing was continued for another 10 minutes. Then 40 μL of DI water was added dropwise to the mixture and then the mixture was ground again for another 5 minutes. The mixture was heated for 60 seconds in an oven at 170 °C affording a reddish solid. However, after copiously washing with DI water and DMSO, only trace solid was obtained.

2. *Maybe I have missed it, but I could not find a solid-state NMR and PXRD data and other electron diffraction data of wCOF-Ag. It should be provided.*

[Reply]: The solid-state NMR of wCOF-Ag (red line) is shown in Supplementary Figure 18. PXRD data of wCOF-Ag (blue line) is shown in Fig. 4b in the manuscript. The single crystal electron diffraction (SCED) data and the experimental and crystallographic parameters of wCOF-Ag were added (Supplementary Figure 20 and Supplementary Table 2).

Supplementary Figure 20 | 3D reciprocal lattice of wCOF-Ag reconstructed from the SCED data. As wCOF-Ag was synthesized by post-synthesis, the local bonding is not very order. It is difficult to directly observe the new formed bonds via SCED data.

Supplementary Table 2 | SCED: Experimental and Crystallographic parameters of wCOF-Ag.

Tilt range (°)	-53.22 ~ 50.17
Tilt step (°)	0.23
Wave length (Å)	0.0251
No. of frames	401
Exposure time per image (s)	0.5
Crystal system	Monoclinic
Possible space group	C2/c, Cc, C2/m, Cm, C2
Unit cell parameters	$a = 15.74 \text{ \AA}, b = 31.11 \text{ \AA}, c = 11.00 \text{ \AA},$ $\alpha = 90^\circ, \beta = 122.6^\circ, \gamma = 90^\circ$
Resolution (Å)	1.15
$I/\sigma(I)$	2.54
Completeness (%)	72.4
CC _{1/2}	95.6
R _{meas} (%)	27.2
No. of total reflections	2775
No. of unique reflections	1145

3. *The IR data of wCOF-Ag should be explained a bit more and so as the PXRD data.*

[Reply]: Thank you very much for your suggestion. As shown in the revised Supplementary Figure 17, we have provided more explanations about the FT-IR data of wCOF-Ag. For PXRD data, Pawley fitting for wCOF-Ag was included in Supplementary Figure 21.

Supplementary Figure 17 | FT-IR spectra showing time evolution of vibrational bands when mCOF-Ag was cross-linked with glyoxal. With longer reaction time, the characteristic peaks of N-H stretches (blue band) and N-H bending vibration (pink band) decrease, indicating the consumption of amine groups, while the peak at 1600 cm⁻¹ (green band) due to newly formed imine bond increases gradually. The red curve corresponds to the control experiment without glyoxal, which was heated in 1,4-dioxane at 70 °C for five days. All KBr pellets were dried at 100 °C before testing to eliminate background signals due to water.

Supplementary Figure 21 | Pawley fitting of wCOF-Ag. Experimental (red), calculated (blue), as well as difference profiles (black) are presented. The position of Bragg reflections under the patterns are shown in green. The refined unit cell parameters are $a = 15.6958 \text{ \AA}$, $b = 29.9200 \text{ \AA}$, $c = 10.6661 \text{ \AA}$, and $\beta = 123.31^\circ$ (space group: $C2$), which are in good agreement with those of mCOF-Ag, indicating the parent structure of mCOF-Ag has been well inherited by wCOF-Ag. The final residuals for the Pawley fitting are $R_{wp} = 0.135$, $R_{exp} = 0.123$.

Response to Reviewer 2:

Reviewer #2 (Remarks to the Author):

This work shows a novel and well-designed synthetic approach to produce single crystals of a 1D-metallo COF using a combination of two different dynamic bonds, coordination bonds and covalent bonds. The structure of the new COF has been successfully solved using single-crystal electron diffraction. Additionally, the initial m-COF-Ag network is further polymerized to form a novel woven 3D network.

The results presented are very important suggesting a new way to obtain 1D COF single crystals in a controlled manner. The manuscript is well-written and deserves publication with very minor corrections.

We thank the reviewer for his/her positive comments.

1. What happen when the metal center is removed from mCOF-Ag and wCOF-Ag?

[Reply]: Thank you very much for your suggestion. To date, the most effective method of removing metal ions is using highly toxic KCN. However, because of safety issue, we could not perform demetalation using KCN in our lab. We have tried to immerse mCOF-Ag in ammonia (*Alternative Demetalation Method for Cu(I)-Phenanthroline-Based Catenanes and Rotaxanes. Org. Lett. 13, 1808–1811 (2011)*) but failed to remove silver ions.

For mCOF-Ag, as Ag ions play an important role in the crystal structure, the removal of silver ions will result in the decomposition of the material into organic chains and pendant amine groups **I**.

In the case of wCOF-Ag, as stated in our procedure of crystalline-state polymerization (Supplementary Materials, page 6), increasing the temperature to 80 °C resulted in the reduction of Ag ions to Ag particles by glyoxal. Therefore, instead of treating with highly toxic KCN, demetalation was successfully carried out by treating wCOF-Ag with a 1,4-dioxane solution of glyoxal at 90 °C. After three days, the obtained material was washed with THF, H₂O, EtOH, and Et₂O sequentially. Subsequent analysis by TEM revealed a large amount of Ag particles on the surface of the material. In addition, the PXRD peaks had also decreased significantly. A strong reduction in PXRD intensity after demetallation was also reported by Liu, Y. *et al.* in *Weaving of organic threads into a crystalline covalent organic framework. Science 351, 365–369 (2016)*.

Figure | PXRD patterns of wCOF-Ag and the Demetalated material. After demetalation, the PXRD intensities decrease significantly. As shown in its TEM image (inset), a large amount of Ag particles are observed on the surface of the material.

2. In order to corroborate the coordination environment of the Ag(I) after the conversion from mCOF-Ag to wCOF-Ag, for instance XPS analysis would be interesting.

[Reply]: Thank you very much for your suggestion. The XPS Ag3d data of mCOF-Ag and wCOF-Ag was added in the Supplementary Materials. The XPS Ag3d spectrum of wCOF-Ag is nearly identical with that of mCOF-Ag, indicating Ag ions in both materials have similar chemical environment.

Supplementary Figure 22 | X-ray photoelectron spectroscopy (XPS) Ag3d data of mCOF-Ag and wCOF-Ag. The XPS Ag3d spectrum of wCOF-Ag is nearly identical with that of mCOF-Ag, indicating that Ag ions in both materials have similar chemical environment.

3. Have the authors analyze the adsorption properties of these new materials?

[Reply]: In the crystal structure of mCOF-Ag, the pores of the framework are occupied by BF_4^- anions. Nitrogen adsorption-desorption experiments revealed that mCOF-Ag showed low uptake in the isotherm line, a characteristic of nonporous materials (as shown in the following figure). Similar results have been reported for other metal containing COFs (*Liu, Y. et al. Weaving of organic threads into a crystalline covalent organic framework. Science 351, 365–369 (2016); Zhao, Y. et al. A Synthetic Route for Crystals of Woven Structures, Uniform Nanocrystals, and Thin Films of Imine Covalent Organic Frameworks. J. Am. Chem. Soc. 139, 13166–13172 (2017)*). Since the mother crystal, mCOF-Ag has low porosity, we didn't analyze the adsorption properties of wCOF-Ag.

Supplementary Figure 7 | Nitrogen adsorption (filled symbols) and desorption (empty symbols) isotherms of mCOF-Ag. The low uptake in the isotherm line is in line with its crystal structure, whereby the pores are occupied by BF_4^- anions.

=====

Response to reviewer 3:

Reviewer #3 (Remarks to the Author):

The manuscript describes the synthesis of a metallated 1-D covalent organic framework - a polymer with a regular connectivity that packs in an ordered solid state structure - with a single crystalline morphology, and its subsequent cross-linking to form a woven network. The initial framework is structurally characterized by single crystal electron diffraction and its non-centrosymmetric structure is confirmed by the observation of non-linear optical properties. Nanoindentation measurements show the enhanced mechanical properties of the woven material. The preparation of single crystalline COFs is rare, and, as the authors printout, particularly so for low-dimensional materials.

As far as I am aware, this is the first report of a 1-D single crystalline COF, albeit a metallated one and that requires electron diffraction, rather than X-rays to fully characterize

the structure. Despite those caveats, the elegance of the synthetic method using reversible chemistry and the metal template to form a highly ordered structure combined with the complexity of the structure make this an article that will be of broad interest to researcher working in the very active areas of framework solids and network materials more widely. The use of templates has obviously been used in diverse areas of chemistry, but given the current interest in preparing highly crystalline COFs, particularly as reports of COFs with varied functional properties are proliferating, methods to produce single or even highly ordered polycrystalline solids are essential. This work adds to those possible tools and facilitating the use of crystallographic methods for COFs will facilitate reliably determination of the structures of COFs in order to better elucidate structure-function relationships in this class of materials.

As an aside, I wonder if this material could have been determined directly from synchrotron XRPD based on an idealized structural model as a starting point.

[Reply]: Thank you for your suggestion. It is difficult to directly determine the fine structure from synchrotron XRPD data alone as the resolution of our data (Supplementary Figure 11) is not good enough, so XRPD has to be used in conjunction with electron diffraction data to confirm the structure.

The manuscript is well written; it is succinct and the text presents the results and supporting evidence very clearly. I would support the publication of this manuscript in Nature Communications.

There are a few minor revisions and comments that I would encourage the authors to address before publication.

We thank the reviewer for his/her positive comments.

1. The main issue is presentational: The figures of the crystal structures are extremely difficult to understand, even viewing them alongside the descriptions in the text. For example, the pi-pi stacking and hydrogen bonding between chains are invoked in the text, but not shown in the figures; the packing diagrams (Fig. 1, 3a) do not show the crystallographic axes and in Fig. 2d, the electron density makes it hard to see the atoms, so overall, it is difficult to visualize the complete 3-D structure without opening the cif in a visualizer. The figures in the ESI do not help significantly. It is very difficult to see how the 'corrugated layers' in Fig. S8

are corrugated and to understand how the other representations of the structure relate to this. I would suggest having a single figure specifically for a simple representation of the structure, even as a schematic, to show some of the higher order features, such as the layers, the strand/chains that make up the structure. The structure itself is beautifully complex and it would be a shame if the authors cannot present it in a clear way.

[Reply]: Thank you very much for raising this. Following your suggestion, we have improved the figures to address these shortfalls. The π - π stacking, hydrogen bond interactions, and the corrugated layer are highlighted in the revised Supplementary Materials, as shown in Supplementary Figures 9 and 10. The crystallographic axes of Fig. 3a was added; as Fig. 1 is a schematic of the strategy, we didn't add the crystallographic axes. The electron density map (Fig. 2d) was used to reveal that all positions of the non-hydrogen atoms (C, N, and Ag) on the framework structure and the locations of the guests (BF_4^- anions) are identified.

Supplementary Figure 9 | Crystal view along the c -axis, in which the $\text{H}\cdots\text{F}$ hydrogen bond interactions between amine and BF_4^- anions are highlighted with a red square. Colour scheme: C on organic chains, blue; C on pendants, grey; N, purple; Ag, yellow; B, pink; F, green; H of amine groups, white.

Supplementary Figure 10 | Tilted side view of the crystal structure, in which a single phenanthroline chain is highlighted in bright yellow. The corrugated layer and the π - π stacking of interlayer phenanthroline rings are also identified. C on organic chains, blue; C on pendants, grey; N, purple; Ag, yellow. Hydrogen atoms and BF_4^- anions are omitted for clarity.

Fig. 3a | The crystal view of mCOF-Ag along the c -axis. Colour scheme: C on organic

chains, blue; C on pendants, grey; N, purple; Ag, yellow. Hydrogen atoms and BF_4^- anions are omitted for clarity.

2. *Do the authors have any idea why is the conjugated polymer crystalline? Is this a fully polymeric material or is there any previous literature that would lead them to expect it to be crystalline? A comment in the ESI would suffice.*

[Reply]: We assume the referee is referring to the low crystalline material named by us as “conjugated polymer” in Fig 1 and Supplementary Figure 1. The semi-crystalline feature of the conjugated polymer is attributed to the rigid and conjugated structure of phenanthroline backbones and the reversible nature of imine bond. There are also previous reports on the semi-crystallinity of imine-linked conjugated polymers (*Fig.S1*, Zhao, Y. *et al. A Synthetic Route for Crystals of Woven Structures, Uniform Nanocrystals, and Thin Films of Imine Covalent Organic Frameworks. J. Am. Chem. Soc.* **139**, 13166–13172 (2017).

We have added this information in the description of Supplementary Fig. 1: “**Supplementary Figure 1** | Experimental PXRD patterns of the conjugated polymer (black) and mCOF-Ag (red). The synthesized conjugated polymer has a semi-crystalline feature, which is attributed to the π - π stacking in the rigid and conjugated structure of phenanthroline backbones; besides, the reversibility in imine bond formation can impart self-correction during the hydrothermal conditions.”

3. *I would encourage the authors to provide a little more information on the treatment of the COF in terms of drying/activation. The framework is washed with DMSO, THF, EtOH. Is it confirmed that no solvent is retained in the structure? The TGA shows significant mass loss before the apparent onset of decomposition at approx. 320 °C - what is being lost?*

[Reply]: We appreciate Reviewer 3 for bringing this to our attention.

To totally remove the solvents adsorbed on the surface of the material, the activation temperature was raised to 120 °C. However, due to the densely packed structure of mCOF-Ag, a small amount of solvents were trapped in the crystal. As shown in the new TGA curve, ~1.4% weight loss below the decomposition temperature was observed.

Supplementary Figure 6 | TGA curve of mCOF-Ag. The decomposition temperature is about 320 °C. The ~1.4% weight loss below 320 °C is attributed to the small amount of solvents trapped in the crystals.

4. I would also suggest a slightly fuller description of the polarized SHG measurement, with in terms of what the experiment actually is and what the results show. It is not a common technique in this area.

[Reply]: Thank you for your suggestion. Polarized SHG microscopy is a feasible solution for characterizing structural morphology and crystallography of various materials. The experimental details for SHG were included in the supporting information. Accordingly, in the revised manuscript (page 4, line 24), we added: “Polarization-dependent SHG was recorded on an isolated crystal. Using polarized laser excitation, a SHG map of the COF-Ag crystal could be obtained (Fig. 3b, inset). The strongest SHG response was observed under parallel polarized excitation, this means that the maximum SHG response originates from the longitudinal direction of the mCOF-Ag crystal.”

5. The authors should present X-ray powder profile simulated from the single crystal cif vs. the experimental XRPD to see how different/similar the bulk microcrystalline powder is from

the determined single crystal structure.

[Reply]: Thank you for your suggestion. The X-ray powder diffraction profile simulated from the determined single crystal structure of mCOF-Ag was presented in Supplementary Figure 12.. The simulated data is highly consistent with the experimental SPXD pattern, indicating the very high quality of the synthesized crystals.

Supplementary Figure 12 | Experimental SPXD pattern and the simulated profile from the single crystal structure of mCOF-Ag. The simulated data is highly consistent with the experimental data, indicating the very high quality of the synthesized crystals.

6. Have the authors tried Rietveld refinement for the woven COF. The apparent similarity of the two materials would suggest the mCOF-Ag would be a reasonable basis for a starting model. Even if Rietveld is not feasible at least Pawley/Le Bail refinement for wCOF-Ag should be included.

[Reply]: Thank you very much for your suggestion. We had tried to conduct Rietveld refinement for wCOF-Ag. However, the direct observation of newly formed bonds was not possible from the SCED data. This is because wCOF-Ag was synthesized via post-synthesis,

thus the local bonding is not very order. Without a very reliable initial structure model, we didn't conduct the Rietveld refinement to interpret the very detailed structural information. Anyway, when comparing the PXRD pattern of wCOF-Ag to that of mCOF-Ag, the very similar patterns between the two materials evidenced that the parent structure (mCOF-Ag) has been inherited by wCOF-Ag after cross-linking. Differences in some of the reflections reveal that their fine structures differ. As suggested by the reviewer, the Pawley fitting of wCOF-Ag was conducted and the result was presented in Supplementary Materials number 21.

Supplementary Figure 21 | Pawley fitting of wCOF-Ag. Experimental (red), calculated (blue), as well as difference profiles (black) are presented. The position of Bragg reflections under the patterns are shown in green. The refined unit cell parameters are $a = 15.6958 \text{ \AA}$, $b = 29.9200 \text{ \AA}$, $c = 10.6661 \text{ \AA}$, and $\beta = 123.31^\circ$ (space group: $C2$), which are in good agreement with those of mCOF-Ag, indicating the parent structure of mCOF-Ag has been well inherited by wCOF-Ag. The final residuals for the Pawley fitting are $R_{wp} = 0.135$, $R_{exp} = 0.123$.

REVIEWERS' COMMENTS:

Reviewer #1 (Remarks to the Author):

I think authors have addressed all comments and this manuscript could be accepted now.

Reviewer #2 (Remarks to the Author):

The revised version is suitable for publication in its current form.

Reviewer #3 (Remarks to the Author):

I am satisfied that the authors have comprehensively responded to the points raised by myself and the other reviewers. I would welcome the publication of the revised manuscript in Nature Communications.